# Norcantharidin Nanoemulsion Development, Characterization, and In Vitro Antiproliferation Effect on B16F1 Melanoma Cells

**DOI:** 10.3390/ph16040501

**Published:** 2023-03-28

**Authors:** Gabriel Martínez-Razo, Patrícia C. Pires, María Lilia Domínguez-López, Francisco Veiga, Armando Vega-López, Ana Cláudia Paiva-Santos

**Affiliations:** 1Laboratorio de Toxicología Ambiental, Escuela Nacional de Ciencias Biológicas, Instituto Politécnico Nacional, Unidad Profesional Zacatenco, Mexico City 07738, Mexico; 2Department of Pharmaceutical Technology, Faculty of Pharmacy of the University of Coimbra, University of Coimbra, Azinhaga Sta. Comba, 3000-548 Coimbra, Portugal; 3LAQV, REQUIMTE, Department of Pharmaceutical Technology, Faculty of Pharmacy of the University of Coimbra, University of Coimbra, Azinhaga Sta. Comba, 3000-548 Coimbra, Portugal; 4Health Sciences Research Centre (CICS-UBI), University of Beira Interior, Av. Infante D. Henrique, 6200-506 Covilhã, Portugal

**Keywords:** cancer, melanoma, nanoemulsion, norcantharidin, skin application, topical administration

## Abstract

Melanoma is a highly lethal type of cancer that has had an increase in incidence in the last decades. Nevertheless, current therapies lack effectiveness and have highly disabling side effects, which calls for new therapeutic strategies. Norcantharidin (NCTD) is an acid derivative with potential antitumor activity isolated from natural blister beetles. However, its solubility limitations restrict its use. To address this issue, we developed an oil-in-water nanoemulsion using commonly available cosmetic ingredients, which increased NCTD solubility 10-fold compared to water. The developed nanoemulsion showed a good droplet size and homogeneity, with adequate pH and viscosity for skin application. In vitro drug release studies showed a sustained release profile, ideal for prolonged therapeutic effects. Accelerated stability studies proved that the formulation was reasonably stable under stress conditions, with particle separation fingerprints, instability index, particle size, and sedimentation velocity analyses being conducted. To assess the therapeutic potential of the developed formulation, in vitro studies were conducted on melanoma B16F1 cells; results showed an IC50 of 1.026 +/− 0.370 mg/kg, and the cells’ metabolic activity decreased after exposure to the NCTD nanoemulsion. Hence, a new “easy-to-make” nanoformulation with therapeutic potential on melanoma cells was developed, as a possible adjuvant for future melanoma treatment.

## 1. Introduction

Although melanoma represents only 4% of cancer diagnoses worldwide, it is responsible for 75% of skin cancer deaths, which speaks for its high lethality [1]. According to GLOBOCAN (Global Cancer Observatory), from the International Agency for Research on Cancer, in 2018 alone, melanoma caused 60,712 deaths. Moreover, it is estimated that 466,914 new cases will be diagnosed by 2040 [2]. The first-line treatment for melanoma is excision surgery, when diagnosed in the initial stages. For late stages, inoperable lesions or lesions with unknown location, immune checkpoint inhibitors targeting cytotoxic T-lymphocyte antigen 4 (CTLA-4) or programmed cell death protein-1 (PD-1) and small molecule BRAF (v-raf murine sarcoma viral oncogene homolog B1) inhibitors are the considered first-line treatment options [3]. Additionally, adjuvant treatments are often used after first-line treatments, to lessen the chance of remission, and may include chemotherapy, radiation therapy, hormone therapy, targeted therapy, or biological therapy [4].

Norcantharidin (NCTD) is a 7-oxabicyclo heptane-2,3-dicarboxylic acid, isolated from natural blister beetles from the Meloidae family (Figure 1A) [5]. NCTD’s action mechanism includes the inhibition of protein phosphatases PP1 and PP2, which play a vital role in cell cycle regulation [6]. Moreover, the anhydride moiety in NCTD has been suggested as an effective alternative to treat several types of cancer, including neuroblastoma, glioblastoma, and melanoma [7]. Despite the shown promise, NCTD’s use is limited due to its low aqueous solubility (maximum of 2.5 mg/mL at pH 6, and 9.5 mg/mL at pH 9.5), which hinders the obtainment of a high drug strength in formulations for potential administration [8]. Hence, nowadays, NCTD’s use is restricted to traditional eastern medicine, in dosage forms such as intravenous and intramuscular injection of its sodium salt in a solution at pH 9. The injection of such an alkaline formulation leads to both local and systemic side-effects, such as intense irritation at the injection site, and irritation of the urinary tract, limiting its clinical application [9].

The skin is an almost impermeable physical barrier that protects the body against external substances that come from the surrounding environment, including therapeutic molecules (Figure 1B). Drug transport through the skin is mainly mediated by transcellular absorption, through the keratin-packed corneocytes, by partition in and out of cell membranes, and by intercellular absorption, around the corneocytes, in the lipid-rich extracellular regions (Figure 1C). The rate-limiting step of a drug’s skin permeation is often its diffusion through the *stratum corneum*, which represents an important challenge in formulation design [10]. Melanocytes are located in the basal cell layer at the deepest part of the epidermis. Hence, an appropriate formulation administered with the intent of making the drug reach these cells should have built-in features that allow active ingredients to reach this area (Figure 1D). Nanosystems, which are vesicular systems with a particle size up to 500 nm, have been a promising strategy for drug delivery and internalization processes, for a grand diversity of administration routes, including topical administration [11]. These systems have many advantages when compared to conventional formulations, such as a controlled drug release, drug absorption enhancement, the improvement of intracellular penetration, an increased retention times at the administration site, and an overall increase in bioavailability at the intended target site [12,13]. Nanoemulsions are colloidal biphasic liquid-in-liquid systems composed of a hydrophilic phase and a hydrophobic phase, where one liquid phase is dispersed as nanosized droplets in the second. They can be classified as oil-in-water (O/W) or, on the contrary, as water-in-oil (W/O) [14]. These systems are stabilized by surfactants, which work at the interface between the two immiscible phases to lower the surface tension and act as a barrier to prevent coalescence driven by the system’s tendency to reach the minimal free Gibbs energy [15]. The hydrophilic–lipophilic balance (HLB) ratio between the used surfactants and their concentration has a high influence on the stability of the system. Clinical evidence increasingly supports the effectiveness of NCTD as an anticancer medication. It can be administered through different routes, including oral, intravenous, and local injection, and can be used alone or in combination with other therapies. Patients with mid- to advanced-stage tumors, such as hepatocellular cancer, esophageal cancer, gastric cancer, lung cancer, ovarian cancer, and non-Hodgkin lymphoma, have experienced reduced tumors, improved quality of life, and prolonged survival time with NCTD treatment [16]. Therefore, it is considered a valuable supplementary medication for the clinical treatment of such tumors and the prevention of postoperative recurrence. The dosage and administration of NCTD vary depending on the route of administration, with oral administration involving a dose of 5–15 mg, intravenous infusion or drip a dose of 10–20 mg, and local injection a dose of 20–40 mg per injection. A well-designed nanoemulsion can have a long shelf life while retaining its original properties through time. This is a very important factor for the translation from bench to a possible clinical application since it can have a significant impact on the outcome of patient treatment. Therefore, it is crucial to characterize the developed nanoemulsions’ physicochemical properties and stability before in vivo experiments, and address scalability issues, in order to ensure optimal development. Furthermore, the selection of common ingredients in dermocosmetic formulations can allow these formulations’ preparation in community pharmacies, which can be a major advantage in what concerns both cost and practical applicability. In the present work, we aimed to develop an O/W nanoemulsion, comprising commonly available ingredients in community pharmacies, to encapsulate NCTD for topical administration as a potential adjuvant therapy for melanoma, emphasizing the obstacles of topical drug delivery and the benefits of using nanoemulsions for overcoming the problems related to this administration route. The general purpose and novelty of the current work resides in the development of a nanoformulation with straightforward preparation methods, with a high drug strength and optimal properties for topical drug delivery, using excipients that are easily accessible for pharmacies worldwide. Hence, one day, it could be easily prepared by pharmacists around the world and be accessible to patients for the adjuvant treatment of melanoma.

## 2. Results and Discussion

### 2.1. Formulation Development

Formulation design is one of the most promising approaches for improving the solubilization, absorption, and consequently, the bioavailability of many drug candidates [17]. A typical “cold cream” is an O/W emulsion in a 30/70 ratio. In our formulation, this proportion was maintained for the required hydrophilic–lipophilic balance needs. The procedure followed in the preparation of the developed nanoemulsion was a standard procedure, quite common in dermatological product development. Moreover, when an active ingredient was present, adding it only once the temperature dropped prevented the degradation of thermolabile compounds. The developed O/W nanoemulsion was designed to include ingredients that are commonly found in dermatological preparations, such as emollients (almond oil), hydrants (urea), lubricants (glyceryl monostearate), humectants (glycerin), and texturizing agents (cetyl alcohol, stearic acid). The selected surfactants were polysorbate 80, a hydrophilic surfactant, and the autoemulsifier Eumulgin B1^®^, a hydrophobic surfactant which provides stability in any polarity condition, a feature that was crucial since our solubility approach was based on a pH interaction.

Sebum is the natural emollient of skin. It provides protection from the external environment by forming a pliable film on its surface. This film consists of squalene, wax esters, triglycerides, cholesterol esters, and free cholesterol. The sebum also provides immunological and antimicrobial protection. Regular washing of the skin removes the sebum, contributing to the development of dry skin [10]. Emollients are ingredients that are commonly present in skin care products, which, when applied to the skin, deposit a lipid film that can also replenish lost skin lipids. Almond oil was selected as an emollient due to its high content in long-chained monounsaturated fatty acids and polyunsaturated fatty acids (PUFAS), mainly oleic (56.6–64.0%) and linoleic (24.5–29.8%) acids [18]. These fatty acids contribute to the skin’s structural integrity and are necessary to synthesize wax esters and squalene. Furthermore, PUFAs such as linoleic, arachidonic, eicosapentaenoic, and docosahexaenoic acids mediate inflammatory and immune responses [19]. The oil’s viscosity is also a crucial parameter for the nanoemulsion formation and obtained droplet size. For instance, low-viscosity oils are used to create droplet sizes ranging from 50 to 100 nm, while nanoemulsions generated with high-viscosity oils, such as long-chain triglycerides, result in much larger droplet sizes than their counterparts [20]. Many emollients also promote the skin penetration of active ingredients by several mechanisms, including an increased solubility of the compound in the stratum corneum, increased partitioning from the vehicle into the membrane, and structural alterations due to the swelling of corneocytes and a rearrangement of the intercellular lipid domains.

Furthermore, to achieve optimal skin responsiveness, humectation and hydration are key factors. The inclusion of glycerin in the water phase modifies the density of water and allows it to remain longer on the skin’s surface before evaporation. Urea, on the other hand, is a small molecule that can easily enter the deeper layers of the skin and retain moisture for a considerable time, allowing the skin to recover from water loss [21]. Moreover, several common skin care ingredients, including glycerol monostearate, have been shown to enhance stratum corneum hydration by a glycerol-dependent mechanism, as the known product of triglyceride hydrolysis in sebaceous glands. Hence, the inclusion of these ingredients can contribute to basic skincare, while also enhancing drug delivery effectiveness [22]. Furthermore, the characteristics of nanoemulsions highly depend on their components, therefore desirable formulation qualities can be acquired by carefully selecting ingredients and determining the amount and weight ratio of oil and surfactants [23].

Nanoemulsions can be prepared by a two-step approach: first a macroemulsion (1–100 µm) is formed; second, the macroemulsion is converted into a nanoemulsion by a high-energy method. By inputting a certain amount of energy into a macroemulsion, high-energy methods can reduce their droplet size and potentially increase their stability. In our work, we used ultrasonication, a highly effective and reliable high-energy method.

### 2.2. Drug Solubility Assays

As mentioned, the potential therapeutic use of NCTD is limited due to its poor solubility in water. Although some attempts have been made to overcome this issue, including the hybridization with poly(lactide-co-glycolide) and galactosylated chitosan, these modifications alter the molecules’ anhydride feature, which could affect their activity moiety [24,25]. Moreover, other approaches use organic solvents, which, albeit effective in increasing drug solubilization, may lead to collateral toxicity and poor stability. The obtained NCTD strength in the developed nanoemulsion was 30 mg/mL, which represented a more than 10-fold increment of NCTD’s solubility compared to water only. The NCTD’s dicarboxylic acid’s inherent trait was crucial to our solubility strategy, performed by using a basic triethanolamine solution, and then a subsequent incorporation into a nanoemulsion. Furthermore, although the safety of triethanolamine has been questioned, the Cosmetic Ingredient Review Expert Panel has concluded on its adequate safety.

To the best of our knowledge, limited attempts have been made to formulate NCTD into nanosystems [26,27], and none for topical administration using different approaches. In a study by Lixin et al., NCTD lipid microspheres were developed for injectable formulations, which showed similar particle size to ours. However, the concentration achieved in their final formulation was only 8 mg/mL, which was a much lower concentration than what we were able to obtain in our study. Moreover, their study was aimed at systemic delivery via intravenous administration, which comes with several disadvantages such as invasiveness, risk of injury, and local and systemic adverse events [28].

### 2.3. Droplet Size, Polydispersity Index, Zeta Potential, pH, and Viscosity

The droplet size, PDI, and zeta potential are the most commonly evaluated formulation parameters for nanoemulsion characterization. In laser diffraction methods, a higher PDI value represents a higher difference in chemical potential between droplets, as a function of the dispersed droplets in the continuous phase. It is a measurement of homogeneity and in nanoemulsions should be typically low, ranging between <0.10 and 0.30, in comparison to macroemulsions (>0.40) [29]. In our study, the droplet mean size and PDI values of the developed nanoemulsion increased as weeks passed by, although no statistical significance was observed (Figure 2). This may be explained by the inherent instability of nanoemulsions and their tendency to evolve into a lower energy state, forming larger droplet sizes and having larger PDI values.

Furthermore, the measured NCTD nanoemulsions’ zeta potential was 0 mV, regardless of the batch preparation’s date and samples’ replicates (results not shown). Although this was expected, since there were no excipients in the formulation that could provide the droplets with charge, it could contribute to the nanoemulsion’s instability, as there was no electric repulsion between the droplets to keep them apart. In general, values either lower than −30 mV or above +30 mV are considered best for a more stable nanosystem. The stability of nanoemulsions is further discussed in Section 3.4.

An additional formulation characterization showed that the developed formula had a pH of 6.5, a value compatible with the skin, and an integrated viscosity of 5489 Pa/s, a value high enough to increase retention at the application site but with a good spreadability. As for the rheological behavior of the developed formulation, the analysis described it as being non-Newtonian, thixotropic, with a yielding effort from 177.7 Pa/s, and a lower limit of 0.4 Pa/s at 361 s^−1^. These results portrayed a nanoemulsion with a proper consistency and good dispersibility, as described by the performance of the emulsion in terms of the input stress (Figure 3A) at a determined shear rate and the rheological behavior (Figure 3B).

### 2.4. Accelerated Stability Assessment

STEP-Technology^®^ experimental data analysis uses terminal formulation creaming velocities to allow the characterization of dispersed droplets size, density, and the extent and type of the distribution. As a near-infrared beam passes through the optical cells containing the samples, it records the migration of particles/droplets, detecting changes across the sample, driven by centrifugal fields. The recorded intensity at every position I(r_i_, t_j_) is normalized by I_0_(r_i_), which corresponds to a measurement in an empty cell. The change in light transmission (T(ri, tj)) shapes the extinction profiles during centrifugation and is calculated according to Equation (1):(1)$$\frac{I\;\left(\mathrm{ri},\;\mathrm{tj}\right)}{I0\left(\mathrm{ri}\right)}=T\left(\mathrm{ri},\;\mathrm{tj}\right)$$

The first recorded profile (red) displays the initial distribution of the sample just after centrifugation starts, and the last recorded profiles (green) illustrate the final distribution of the particles/droplets over sample height (position r), due to separation (Figure 4). Depending on the type of dispersion, particles/droplets will show a tendency to sediment (position of the meniscus) or to form a cream (position of the cell bottom) [30]. As expected, the developed NCTD nanoemulsion formed a cream, since it is the common destabilization process for emulsions.

Aside from the extinction profiles of the different nanoemulsion batches, the analysis also provided several parameters relevant to the formulation’s stability, such as the droplet size, creaming velocity, and instability index.

Given the fact that sample cells are placed horizontally inside the equipment, for a monodisperse sample, all profiles have a vertical transmission profile. Therefore, all particles/droplets move the same distance (h = r − rm), and a single profile is obtained. For polydisperse systems, the profiles deviate from the vertical shape and become more and more skewed over time (t_i_). This results in a display of the different traveling distances from the smaller to the larger particles [31]. NCTD nanoemulsions showed a relatively polydisperse profile for all batches, regardless of the preparation date (Figure 5).

By spatial resolving techniques, creaming velocity (v) is determined by measuring the required time (t) for droplets to settle from the meniscus (rm) to a fixed radial position (r, analytical centrifugation), as denoted by Equation (2):(2)$$v\left(r,\;t\right)=\frac{r-\mathrm{rm}}{t}$$

Particle traceability and a high accuracy can be achieved for the determination of creaming velocity. Contrary to light-scattering methods (laser diffraction), which use the light intensity patterns to deduce the particle size based on applied algorithms, in photocentrifugation, light intensity is simply recorded to determine the distance that the particles travel over a time interval and is not used to determine the particle size.

A summary of the information regarding the accelerated stability characterization of the developed NCTD nanoemulsion is displayed in Table 1.

As observed in Table 1, the droplet size increased with time. Accordingly, the creaming velocity decreased with time, since bigger droplets tend to move more slowly than smaller ones. Furthermore, a discrepancy in the determined droplet size values was found between the two different employed methods, which could be explained by the inherent differences between light scattering and photocentrifugation. In the determination of the creaming velocity, the variables were time and distance, and no assumptions such as density, viscosity, refractive index, shape, and empirical models or fitting parameters were made. Additionally, unlike light-scattering methods, the determination of polydispersion was qualitative, rather than quantitative (index).

Nanoemulsions are kinetically stable, and in time, phase separation will occur through different mechanisms. An instability index is measured from 0 to 1.0, with 1.0 being the value for which the formulation is less stable. In our results, the less stable nanoemulsion was the most recent batch (1 w.o.), followed by 3 w.o., and 4 w.o. This might be explained by the fact that the amount of energy that was given to the system by ultrasonication, which was quite high, tended to make the system reverse to a lower energy state as time passed, consequently being more stable. However, the assessment of stability for longer periods of time would be desirable to fully understand this tendency in formulation performance.

### 2.5. In Vitro Drug Release

Nanoemulsions can have unique advantages for topical administration, and many studies have focused on using nanoemulsions for topical drug delivery [32,33,34]. However, to the best of our knowledge there is no information regarding NCTD’s release on topical nanoemulsion formulations. Furthermore, although in vitro drug release assays do not directly measure bioavailability, they may help predict the formulation’s performance, before transitioning to early animal and therapeutic models’ studies.

Our results showed that the most recent nanoemulsion batch (1 w.o.) had a larger cumulative drug release, reaching up to 59.4%, followed by the 3 w.o. and the 4 w.o. batches, with 53.8% and 47.8%, respectively (Figure 6). This could indicate that the formulation loses drug release capacity over time, but a multiple regression analysis revealed no significant differences between batches, and hence NCTD’s release could be considered as being consistent. The formulation showed a sustained release, which is ideal for topical drug delivery (Figure 6 and Table 2).

In cancer treatment, primary therapies often have devastating side effects. A topical neoadjuvant may benefit the course of the ongoing treatment. Neoadjuvant therapies have been reported to sensitize the tumor, lead to immunological activation, and reduce toxicity. Currently, several topical formulations are employed for nonmelanoma treatment, such as 5-fluorouracil (5-FU) cream, 5-aminolevulinic acid (photodynamic therapy), and ingenol mebutate. Mechanistically, these drugs could treat melanoma. However, their limited penetration through the stratum corneum can significantly decrease their potency [35].

### 2.6. Melanoma Cells Nanoemulsion Exposure Assays

NCTD has been proposed for many pharmacological purposes but mainly as a chemotherapeutic agent. However, its use in humans has not been well established as, to the best of to our knowledge, no clinical trial has been done for any therapeutic application. To assess the developed formulation’s antimelanoma potential, B16F1 melanoma cells were exposed to it, at a dose defined by the maximum concentration of NCTD detected in the in vitro drug release assay (16 mg/kg). Two other drug doses were used, the amount of drug detected at 20 min in those same release assays (6 mg/kg) and the systemically safe dose of 3 mg/kg. In a previous study, we evaluated the toxicity profile of NCTD in an in vivo murine model. We concluded that NCTD had a relatively low toxicity profile and that doses up to 10 mg/kg/day could be used safely for up to 10 days. However, the study also found that higher doses of NCTD (20 mg/kg/day) led to signs of toxicity, including weight loss and histopathological changes in the liver and kidneys [36]. Therefore, the use of NCTD in a nanoformulation could help to reduce skin damage, which is often associated with the use of NCTD, and allow for longer periods of NCTD use, if the systemic safety dose is not exceeded. It is important to carefully monitor the dose of NCTD and to consider the potential toxic effects of higher doses when using NCTD for therapeutic purposes.

Microscopically, these cells’ morphology before treatment was fusiform, as depicted in the control group micrography (Figure 7A). The results showed that, as the concentration of NCTD increased, a change in cell morphology was observable, as they became round and small and predominantly dead. These changes were in line with the cell assay analysis, since cell mortality increased at higher drug doses, with an IC_50_ calculated as 1.026 +/− 0.370 mg/L (Figure 7).

Previous studies have shown that norcantharidin suppresses tumor growth in different cell lines and in in vivo models. Mei et al. (2018) investigated the effect of NCTD on human osteosarcoma cells and found that it could inhibit cell proliferation and induce apoptosis in a dose-dependent manner. They also identified the c-Met/Akt/mTOR signaling pathway as a key mediator of NCTD’s antitumor effects. In addition, the authors observed that NCTD treatment led to a reduction in the expression of Bcl-2, an antiapoptotic protein, and an increase in the expression of Bax, a proapoptotic protein, suggesting that NCTD induced apoptosis via the intrinsic pathway [37]. Another study, by Wang et al., investigated the effect of NCTD on tumor growth and metastasis in gastric cancer. The authors found that NCTD could suppress tumor growth and inhibit metastasis by downregulating the expression of MMP-2, which is involved in the invasion and metastasis of cancer cells. The authors also observed that NCTD treatment led to a decrease in the activity of NF-κB, a transcription factor involved in regulating MMP-2 expression [38]. These studies support the potential therapeutic value of NCTD in cancer treatment, particularly in inhibiting tumor growth and metastasis. However, further research is needed to fully understand the underlying mechanisms of NCTD’s anticancer effects and its potential use in clinical settings.

Cell metabolic activity can be determined by measuring NADH and NADPH contents, as these pyridine nucleotides are produced as metabolic activity byproducts. The direct quantification of these reducing agents is feasible, but the quantification of their turnover rate is more meaningful. The turnover rate can be evaluated by a selective reduction of tetrazolium salts or resazurin, since the enzymatic reduction of these compounds by dehydrogenases uses NADH/NADPH as co-substrate [39]. Resazurin is reduced to resorufin by the aerobic respiration of metabolically active cells. Thus, this redox dye is used as an indicator of active metabolism in cell cultures [40]. In general, a significant reduction in the metabolic activity of B16F1 cells was found after exposure to NCTD (Figure 8).

A prior investigation assessing the toxicity of NCTD revealed liver and renal damage, as well as hypoglycemia and cholesterol dysregulation in blood serum samples. Additionally, systemic inflammation was observed, which suggests that NCTD may trigger an immune response and cause inflammation. These metabolic changes may indicate that NCTD interferes with glucose and cholesterol metabolism. Furthermore, the increased activity of enzymes related to NCTD’s binding activity sites and liver function, such as protein phosphatase-1 and alanine transaminase, respectively, may suggest that NCTD can cause reversible hepatic cell damage. Noteworthy, the dose-dependent decrease in reactive oxygen species (ROS) observed in the study is also interesting. ROS play a crucial role in several cellular processes, such as protein phosphorylation, the activation of transcriptional factors, apoptosis, immunity, and differentiation. A decrease in ROS production could therefore have a significant impact on cellular function and potentially contribute to the observed toxicity of NCTD [37]. Nevertheless, drug incorporation into nanosystems is known to reduce their toxicity, and since the developed formulation is meant to be administered topically, this drug’s adverse events are bound to be significantly reduced upon administration.

Surgery is still the first-line treatment for melanoma and the only treatment performed for patients with stages I to III. The fundamental objective of this procedure is the removal of both visible and microscopic tumors and microscopic and macroscopic satellites [41]. Adjuvant therapies have been developed and proved effective in other types of cancer, such as breast and lung cancer, and may include chemotherapy, immunotherapy, radiation therapy, and targeted therapies. However, mainly two approaches have been adopted in recent years: anti PDL-1 and/or BRAF inhibitors [42]. Wang L. et al. evaluated the in vitro effectiveness of various antitumor agents as adjuvants, including NCTD, against vemurafenib-resistant melanoma cells (A375R) in comparison to their parental cells (A375). In A375 cells, the IC_50_ value of vemurafenib was 0.157 μM, whereas A375R cells had an IC50 value of 21.5 μM, which was 100-fold larger than that of A375 cells. Hence, NCTD exhibited a significantly greater cytotoxicity against A375R cells than A375 cells. Flow cytometry analysis indicated that NCTD induced cell cycle arrest in the G2/M phase in A375R cells, with increased Cyclin B1 expression and decreased c-Myc, HIF-1α, and Cyclin D1 levels. The study revealed that NCTD had a more significant inhibitory effect on proliferation in A375R cells than in A375 cells, implying that A375R cells were more responsive to NCTD than A375 cells [43].

## 3. Materials and Methods

### 3.1. Materials

Almond oil, Carbopol^®^ 940, cetyl alcohol, glycerin, glyceryl monostearate, methylparaben, propylparaben, stearic acid, triethanolamine, Tween^®^ 80 (polysorbate 80), and urea were all provided by Merck (Burlington, MA, USA). Eumulgin^®^ B1 (Ceteareth-12) was provided by BASF (Ludwigshafen, Germany). NCTD was provided by Sigma-Aldrich (St. Louis, MO, USA). Ultrapure Milli-Q water was obtained from the purification system Ellix Essential Millipore^®^ (Darmstadt, Germany).

### 3.2. Formulation Development

The formulation was designed according to the HLB requirements, in correspondence to the proportion of ingredients in the oil phase. Starting from a common “cold cream” emulsion, the oil phase consisted of almond oil (0.7 g) as an emollient, cetyl alcohol (0.2 g) as an emulsifier and thickening agent, ceteareth-12 wax (0.2 g) as a hydrophobic surfactant, stearic acid (0.3 g) for texture, and glyceryl monostearate (0.2 g) as a lubricant. The water phase was elaborated with glycerin (0.6 g) as a humectant, urea (0.5 g) as a hydrant, polysorbate 80 (0.05 g) as a hydrophilic surfactant, Carbopol ^®^ 940 (0.01 g) as a rheological modifier, and Milli-Q water (q.s.). In what concerns the production method, both phases were heated at 70 °C and mixed under constant stirring for one minute. After cooling for some time, the mixture was maintained at 40 °C, and a solution of the drug dissolved in 1 mL of triethanolamine-water (20% triethanolamine) was added to the emulsion. The resulting mixture was stirred for an additional minute. The final drug concentration was 3.0% *w*/*w* NCTD (30 mg/mL). The resultant (macro)emulsion was sonicated for 5 min with an ultrasonic processor at an output of 5.0 watts (Vibracell, Sonics & Materials, Newton, MA, USA), in order to obtain a nanometric droplet size.

### 3.3. Drug Solubility Assays

The maximum solubility of NCTD in a triethanolamine-water solution (20% triethanolamine) was assessed by the saturation shake-flask procedure [44]. This procedure consists of evaluating the solubilization of a drug in a specific dissolving agent by constantly stirring increasing concentrations of the solute until it ceases to dissolve and, therefore, turbidity is observed. Hence, fixed NCTD quantities of 25, 50, 100, 200, 300, and 400 mg were added to 1.0 mL of 20% triethanolamine and vortexed until visible saturation occurred.

Aside from assessing drug solubility in the 20% triethanolamine solution, the drug concentration in the final nanoemulsion was also determined, but this time by quantifying it with fluorescence spectroscopy (Synergy MX, Biotek, Urbana, IL, USA), in 96-well plates (Thermo Fisher Scientific, Waltham, MA, USA), with an excitation/emission wavelengths of 360/40 and 480/20 nm, respectively, and calculating it by a regression analysis. The calibration curve was performed by fortifying the nanoemulsion vehicle (without the drug) with fixed quantities of NCTD (10, 12, 14, 16, 18, 20, and 40 mg) dissolved in a solution of 20% triethanolamine in water as standards (Equation (3)):y = 0.529x − 3.451; R^2^ = 0.991(3)

For further analysis, batches with different preparation dates: 1 week (1 w.o.), 3 weeks (3 w.o.) and 4 weeks (4 W.o.) old were compared, in triplicate.

### 3.4. Droplet Size, Polydispersity Index, Zeta Potential, pH, and Viscosity

The characterization of the droplet mean size and polydispersity index (PDI) was done by dynamic light scattering, and the zeta potential was determined by electrophoretic light scattering, using a Zetasizer Nano Series ZS apparatus (Malvern Panalytical, Malvern, UK). Samples were diluted (1:40 *v*/*v*) in Milli-Q water and measured at 25 °C in disposable polymethyl methacrylate cuvettes (DTSS0012) for their droplet size and PDI, and a folded capillary cell (DTS1070) (Malvern, UK) for the zeta potential.

The pH of the nanoemulsion was also measured, using a pH-meter (AB315, Fisher Scientific).

Rheological characterization was performed using a controlled stress rotational rheometer (AR G2, TA Instruments). The used geometry was a smooth 316-grade stainless steel circular parallel plate, with a 60 mm radius (Plate SST ST). Firstly, samples were placed in a Peltier plate, covered by filter paper #150, to prevent fluid displacement. Then, measurements were performed with ascending and descending stress sweep cycles, a log mode, and with a flow ramp from 10 to 150 Pa, with 30 measured points per decade and 15 s intervals between measurements, with a total duration of 675 s. Measurements were performed at 25 °C in a 1 mm gap [45].

### 3.5. Accelerated Stability Assessment

NCTD nanoemulsion samples (500 µL) were diluted 1:1 in Milli-Q water and put inside capped disposable polycarbonate cells, with a pad length of 2 mm. Space- and time-resolved extinction profiles (STEP-Technology^®^) were determined by photocentrifugation at 4000 rpm, for 30 min and at 25 °C, at a wavelength of 865 nm (CCD-line detector). Transmission profiles were recorded in 5 s intervals, for a total of 350 profiles (LUMiSizer LUM GmbH, Berlin, Germany). The apparatus allowed the determination of he sample’s creaming velocity, instability indexes and droplet size, which were determined using the equipment software (LUMiSizer SEPView^®^, Berlin, Germany) [30].

### 3.6. In Vitro Drug Release Assay

In vitro drug release was determined in a Franz Diffusion Cell System (SES GmbH, Waldbrunn, Germany). Receptor compartments were filled with 5 mL of phosphate buffer pH 5.6 (skin pH), and dialysis tubing cellulose membranes (Sigma-Aldrich, St. Louis, MO, USA) were damped in buffer for 30 min before the experiments and then assembled along with the donor compartment. With each chamber mounted, the temperature was set at 36 °C (mean skin temperature) and left to stabilize for 1 h, with constant stirring at 3000 rpm (magnetic stirrer bars). Then, 1 g of the different NCTD nanoemulsion batches (1, 3, or 4 w.o.) was placed in the donor compartment. A volume of 200 µL was collected from the bottom compartment at specific time points, 20, 40, 60, 120, 180, and 240 min, and replaced with fresh buffer. Samples were then quantified by absorbance reading at 480 nm, in 96-well plates (spectrophotometry, Synergy MX apparatus, BioTek, Shoreline, WC, USA). To assess NCTD concentrations in the collected samples, a calibration curve was done priorly, consisting of solutions of NCTD at different concentrations (1.5, 3, 4.5, 6, and 7.5 mg/mL) dissolved in buffer, and using that same buffer as blank. Absorbance values were transformed by squaring, and a second-order equation was obtained (Equation (4)):
(4)$$Y=3e^{-6}x^{2}-2e^{-5}x+4e^{-5};\;R^{2}=0.996$$

The same transformation was performed on the data from samples, and the concentration was predicted by the derivation of the calibration curve (Equation (4)) by the Po Shen Lo method, after outliers’ removal [46]. Data are presented as the cumulative drug release percentage and drug release rate, with the drug release rate being calculated by a data transformation (dividing the drug release percentage by the membrane area, 0.78 cm^2^, and by calculating the square root of time).

### 3.7. Melanoma Cells Nanoemulsion Exposure Assays

Melanoma B16F1 immortalized cells were acquired from the American Type Culture Collection (ATCC) and cultured using the DMEM (Dulbecco’s Modified Eagle Medium) nutrient mix, supplemented with 5% bovine fetal serum, 2 mM glutamine, 1% antibiotics (10,000 U/mL erythromycin, 10,000 U/mL of ampicillin) and pyruvate (all from Life Technologies, Carlsbad, CA, USA). The incubation was conducted in a humidified incubator, at 37 °C, with 5% CO_2_ and 95% air atmosphere saturation [47]. Cells were then placed in 24-well plates, at a cellular density of 250,000 cells/cm^2^. Afterwards, the developed nanoemulsion, with fixed NCTD concentrations of 3, 6, and 16 mg/mL, was added to the medium. Samples were collected 24 h after exposure, and cells were stained with 1.0 mL of a 0.4% sulforhodamine B dye solution and maintained for 10 min at room temperature. The exceeding dye was discarded and washed with 500 µL of a 1.0% acetic acid solution. Finally, 1.0 mL of 10 mM TRIS solution (pH 10.5) was added, and the dye uptake was measured by absorbance (Abs) reading at 550 nm (spectrophotometry, Synergy MX microplate reader, BioTek) [48]. The half-maximal inhibitory concentration (IC_50_) was predicted for cell mortality, calculated according to Equation (5) for colorimetric assays [39]:(5)$$\begin{array}{c} \%\;\mathrm{Cell}\;\mathrm{viability}=\;\frac{{\mathrm{Abs}}_{\mathrm{Samples}\;}-{\;\mathrm{Abs}}_{\mathrm{Blank}}}{{\mathrm{Abs}}_{\mathrm{Control}\;}-{\;\mathrm{Abs}}_{\mathrm{CtrlBlk}}\;}\;\times \;100.\\ \%\;\mathrm{Mortality}=100-\%\;\mathrm{Cell}\;\mathrm{viability}. \end{array}$$

Mitochondrial activity was also tested, with the same cultured B16F1 cells, at a cellular density of 50,000 cells/cm^2^, being placed in a 96-well plate and exposed to the same fixed concentrations of NCTD. After a 24 h exposure, cells were stained with 10 µL of a 1.0% resazurin phosphate buffer solution for 4 h. Samples were then measured for fluorescence at an excitation/emission wavelength of 530 and 590 nm, respectively, and corrected against a blank (Synergy MX microplate reader, BioTek). Mitochondrial stress was calculated as a function of the reduced resazurin molar extinction coefficient. Data are presented as mol/min per 50,000 cells [49].

### 3.8. Data Analysis

All data were evaluated for their goodness of fit to a normal distribution with an Anderson–Darling test and contrasted using an ANOVA, with a Tukey or Kruskal–Wallis post hoc test, accordingly. Regression models assessing for linearity and fit of the prediction values (Equations (3) and (4)) were evaluated in triplicate and verified by checking residual plots. Data from rheological analysis are presented as the nanoemulsion’s viscosity (η) flow curve (ascending and descending), plotted as a function of stress (σ; Pa) and shear rate (γ; 1/s) using the equipment’s integrated software. To obtain a viscosity integrated value, taking data from the flow curve plot, the area under the curve (AUC) from the descending curve was subtracted from the AUC of the ascending curve. Data from the accelerated stability studies, such as droplet size and creaming velocity, were analyzed using an F test. Drug release cumulative percentages and rates were compared for statistical significance using a multiple linear regression model and contrasted using an ANOVA test [32]. Statistical comparisons were performed using PAST^®^ version 2.17c [50]. IC_50_ was calculated using R for Statistical Computing (R: Language and Environment for Statistical Computing 2022) using a Log-logistic model, fitted using 2 parameters: IC_50_ and cell mortality [51,52]. Data curation and plots were created with Excel (Microsoft Corporation, 2018, Redmond, WC, USA) and Sigma Plot^®^ (Version 11.0 Systat Software, 2008).

## 4. Conclusions

Nanoemulsions are versatile drug delivery systems that utilize their amphiphilic nature to solubilize water-insoluble drugs and offer various strategies to overcome solubility limitations. Our study successfully incorporated a significant amount of NCTD into a nanoemulsion formulation by leveraging the acidic nature of NCTD and the basic pH of triethanolamine solution. The formulation had a good dispersibility and adequate surface coverage, and the droplet size was uniform and in the nanoscale range. The formulation showed a controlled drug-release profile, making it appropriate for topical administration. The formulation was also reasonably stable, allowing for reproducibility and quality control. While in vitro experiments demonstrated a proof of concept by affecting the proliferation and metabolic activity of melanoma B16F1 cells, further in vivo and clinical studies are necessary to fully assess the efficacy and safety of the developed NCTD nanoemulsion. Hence, we proposed a NCTD nanoemulsion as a viable adjuvant in the treatment of melanoma. Neoadjuvant therapies such as this could be quite promising for diminishing treatment relapse and increasing the response to first-line treatments.

## Figures and Tables

**Figure 1 pharmaceuticals-16-00501-f001:**
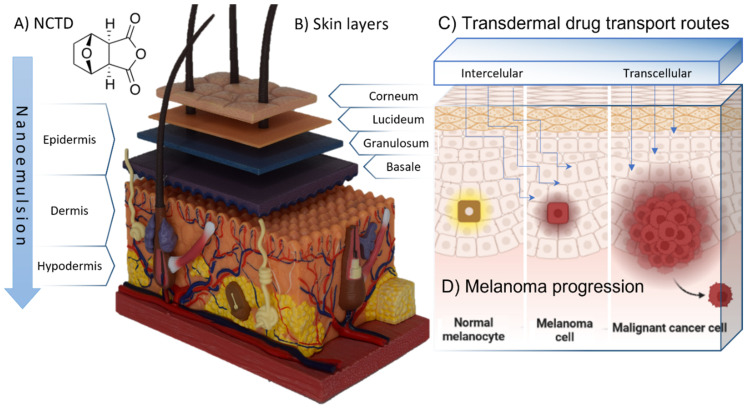
(**A**) Chemical structure of norcantharidin (NCTD) molecule. (**B**) Representation of the skin layers and epidermal strata (skin anatomy model, 4D human anatomy Fame Master^®^ photography). (**C**) Transdermal formulation drug transport routes (created with BioRender^®^). (**D**) Schematic representation of melanoma progression (created with BioRender^®^).

**Figure 2 pharmaceuticals-16-00501-f002:**
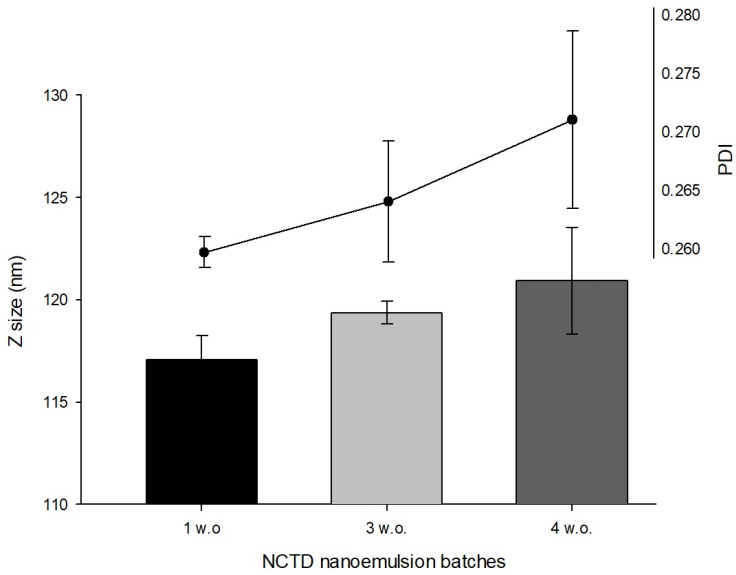
Droplet size (Z size) and polydispersity index (PDI) of NCTD nanoemulsion batches.

**Figure 3 pharmaceuticals-16-00501-f003:**
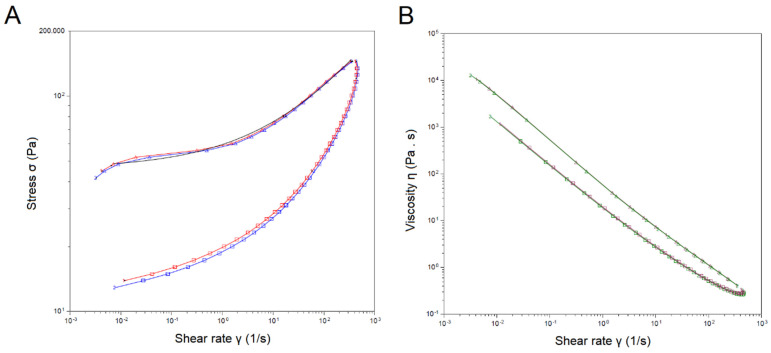
Rheological analysis of the developed NCTD nanoemulsion. (**A**) Flow rate as described by the stress input (σ; Pa) and shear rate (1/s) (red and blue lines correspond to two different batches). (**B**) Viscosity (Pa/s) as a function of shear rate (green and purple corresponding to two different batches). Calculated yield stress of 45.1 Pa, rate index of 0.345, and R^2^ of 0.997.

**Figure 4 pharmaceuticals-16-00501-f004:**
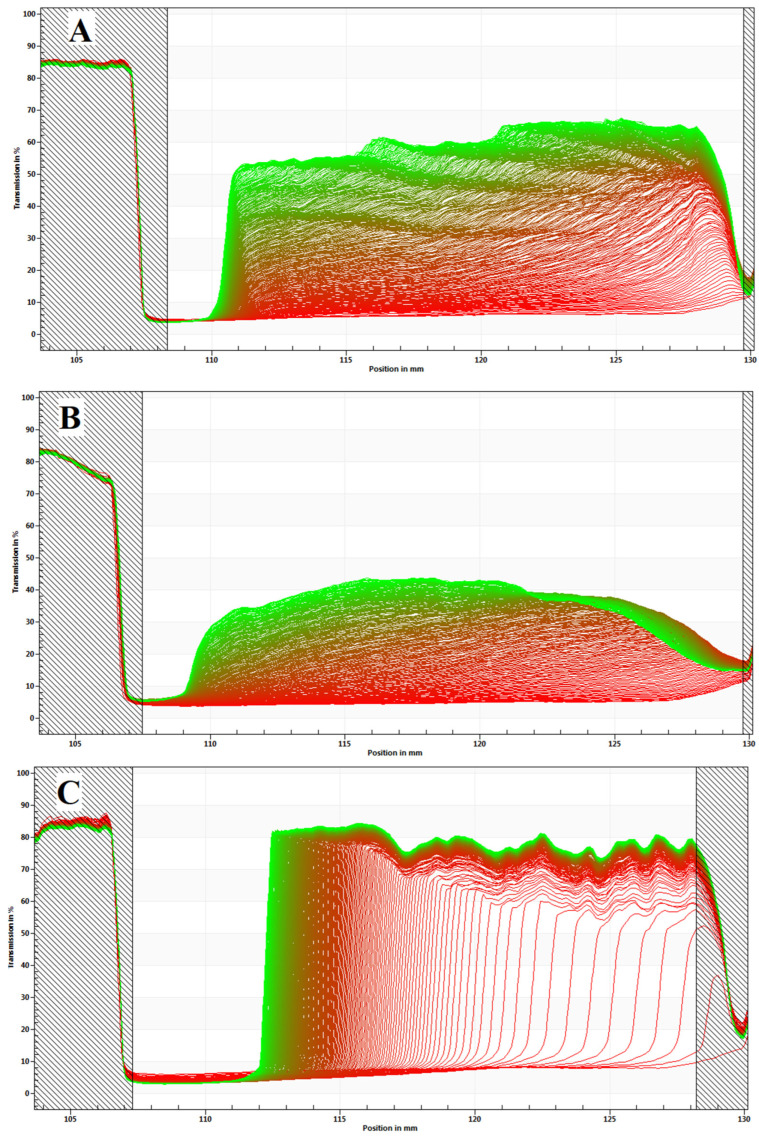
Space- and time-resolved extinction profiles of NCTD O/W nanoemulsions, from the different batches: (**A**) 4 w.o., (**B**) 3 w.o., and (**C**) 1 w.o. Recorded position (I, nm) versus transmission shape (T, %). Creaming formation can be observed in all batches, characterized by the curve formed.

**Figure 5 pharmaceuticals-16-00501-f005:**
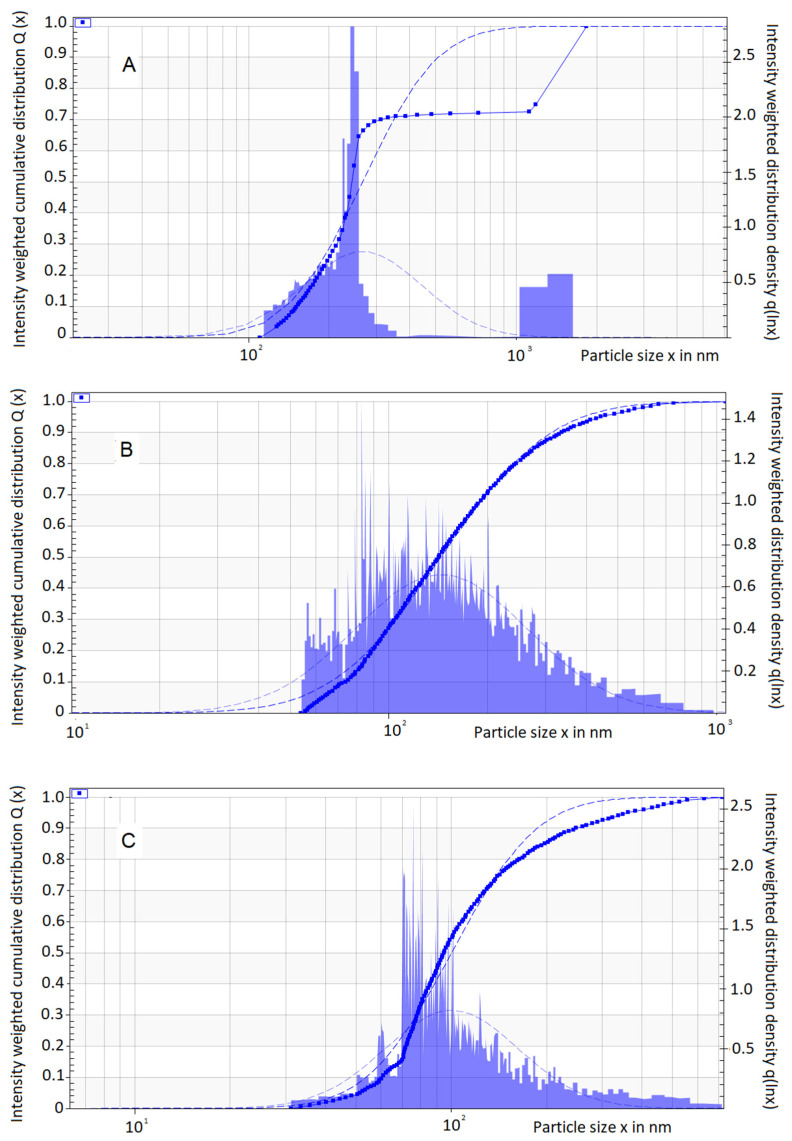
Graphical representation of droplet (particle) size distribution as a function of velocity, for batches prepared at different times: (**A**) 4 w.o., (**B**) 3 w.o., and (**C**) 1 w.o. Intensity-weighted distribution density (right vertical axis), intensity-weighted cumulative distribution (left vertical axis), and particle size (horizontal axis) are represented.

**Figure 6 pharmaceuticals-16-00501-f006:**
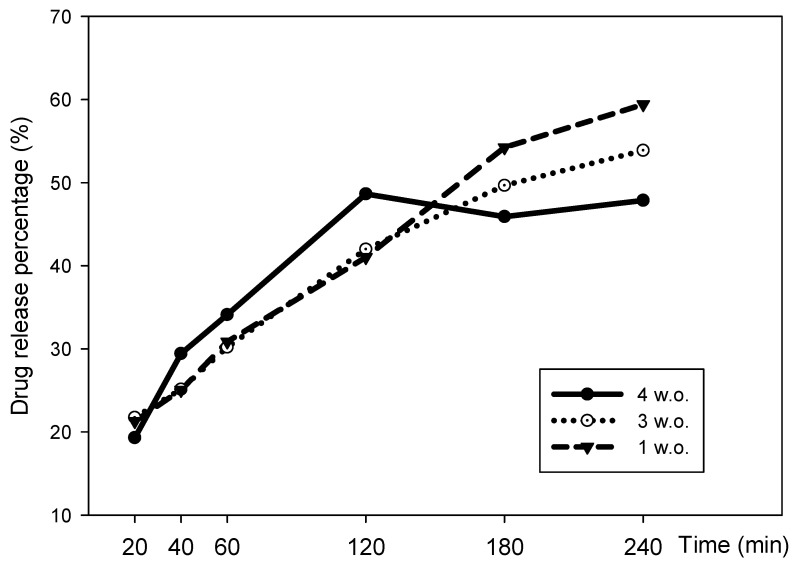
Drug release percentage of NCTD nanoemulsion batches, 1, 3, and 4 w.o.

**Figure 7 pharmaceuticals-16-00501-f007:**
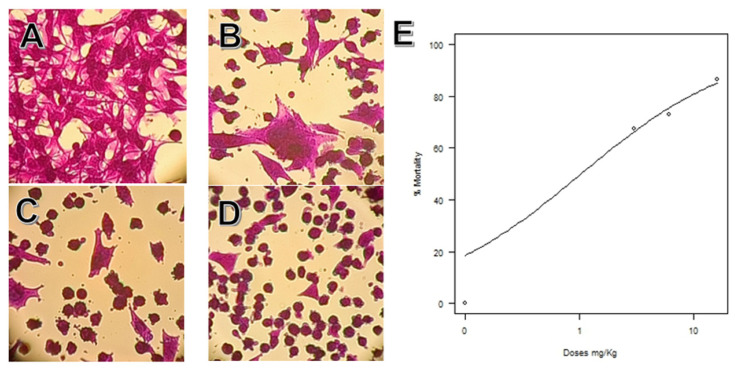
Sulforhodamine’s cell growth test of B16F1 cells exposed to different doses of NCTD. Micrographs (40×) of (**A**) control group, (**B**) 3 mg/kg, (**C**) 6 mg/kg, (**D**) 16 mg/kg; (**E**) graphic representation of the Log-logistic model fit of the IC50 and % of cell mortality. Plot data: For slope: −0.637, S.E.: 0.135, t-value: −4.69 (*p* < 0.001); and for the IC50 intersection: 1.026, S.E. 0.370, t-value 2.76 *; residual std error 6.44 and 22 degrees of freedom (*p* < 0.01).

**Figure 8 pharmaceuticals-16-00501-f008:**
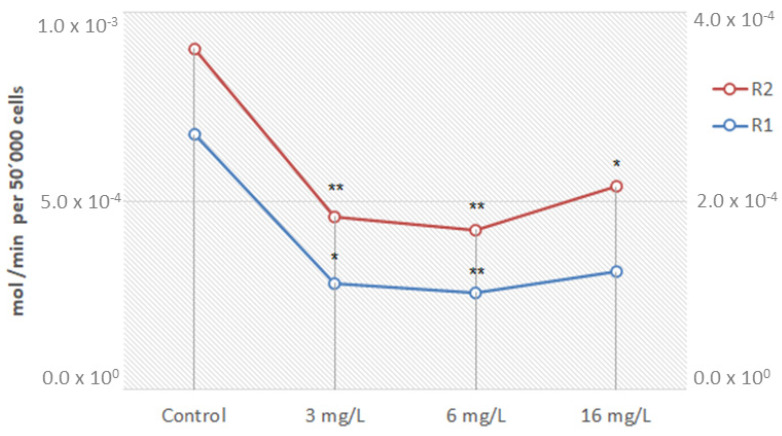
Metabolic effect in B16F1 cells after norcantharidin exposure. Data are presented as resazurin oxidized (mol/min) per 50,000 cells in duplicate (R1 right vertical axis and R2 left vertical axis; ANOVA *p* (same) 1.67 × 10^−4^ and 5.4 × 10^−3^, respectively). * *p* < 0.05; ** *p* < 0.01 compared to the control group.

**Table 1 pharmaceuticals-16-00501-t001:** Accelerated stability characterization parameters of the developed NCTD nanoemulsion, for different batches.

Batch	Droplet Size (nm)			Creaming Velocity (µm/s)			Instability Index	
Id	Median	S.D.	sig	Median	S.D	sig	Mean	sig
4 w.o.	243.6	575.6	-	8.42	59.63	-	0.321	-
3 w.o.	142.1	59.63	^a **^	14.5	82.28	^a **^	0.568	-
1 w.o.	94.7	104	^a ** b **^	53.6	77.9	^a ** b **^	0.658	-

Data are presented as the median for the particle size and the harmonic mean for the creaming velocity and instability index. Statistical significance (sig) is expressed as “a” for the 4 w.o. batch, and “b” for the 3 w.o. batch; ** *p* < 0.01.

**Table 2 pharmaceuticals-16-00501-t002:** Drug release rate and cumulative drug release rate’s multiple linear regression analysis of NCTD nanoemulsion batches.

	Mean	S.D.	Coeff.	Std. Err.	*p*	*R* ^2^
DRR (%·cm^2^·t^1/2^)						
4.w.o.	48.14	13.94	0.070	0.055	0.336	0.845
3 w.o.	47.56	15.61	−0.079	0.218	0.753	0.992
1 w.o.	49.53	18.33	0.232	0.153	0.269	0.989
CDRR (mg·cm^2^·t^1/2^)						
4.w.o.	24.34	13.31	0.417	0.300	0.299	0.994
3 w.o.	23.66	13.57	−0.296	1.173	0.824	0.979
1 w.o.	24.22	14.45	0.163	0.857	0.866	0.971

DRR: drug release rate; CDRR: cumulative drug release rate. Data are presented as the mean of each variable +/− standard deviation (S.D.); and as coefficients (Coeff.) with standard error (Std. Err.) for ANOVA. Statistical values for DRR were F: 179.44, multiple R2: 0.996, and multiple R2 Adj: 0.990 (*p*: 0.005). As for CDRR, the values were F: 118.99; multiple R2: 0.994, and multiple R2 Adj: 0.986 (*p*: 0.008).

## Data Availability

Data is contained within the article.

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
