# Peer review of "Norcantharidin Nanoemulsion Development, Characterization, and In Vitro Antiproliferation Effect on B16F1 Melanoma Cells"

_pharmaceuticals, 2023, doi:10.3390/ph16040501_

Round 1

Reviewer 1 Report

Dear Authors,

It is recommended that a clear description of its novelty be added to the introduction section.

Author Response

Thank you for taking the time to review our work and provide your valuable feedback. We greatly appreciate your suggestion to provide a more clear description of the novelty of our research in the introduction section. Hence, we have revised the introduction section to include a detailed and comprehensive description of the novelty of our work, from lines 110 to 119 (marked in yellow).

Reviewer 2 Report

This manuscript investigated the role of  norcantharidin nanoemulsion on melanoma cells. This article claims that using of nanoemulsion of norcantharidin  could be a suitable for biomedical applications, as well as potential adjuvant therapy for topical melanoma treatment. Therefore, I suggest a minor correction and require a detailed clarification. Correction to be addressed by the authors as follows: The abstract is not well organized, where the sentences are incomplete and no continuity is there. It would be feasible, if include the significance of the current study in the abstract. A brief description of how the authors selected information from the literature in the databases, as well as doses.
Authors should justify and expand the information on the biomedical application of nanoemulsion system in which Norcantharidinis mentioned, highlighting the main contribution in in vitro fields.

Authors should specify the main experimental conditions used on the evidences from the literature. Where they briefly describe the most important data reported in the literature in a homogeneous manner and sequence reinforcing the relevance of this system as medicinal alternative.
The most significant  mechanism of action of this nanoparticles should be described and noticed more emphatically. Authors should discuss whether the use of these nanoparticles represents a solid alternative to existing commercial drugs or a source of new drugs.
Please add below studies to your manuscript in discussion section and also please discuss about possible toxicity of proposed nanomaterials and mitochondria targeting using this system.

DOI: 10.3389/fphar.2022.906043

DOI:10.1016/j.trac.2019.05.004

DOI: 10.1186/s13020-020-00338-6

Conclusions should reaffirm the fundamental contribution of this paper.

Author Response

Thank you for your review and feedback on our manuscript. We appreciate your comments and suggestions, which have helped us improve the quality of our work. A point-by-point answer is given bellow, and changes have been marked in the revised version of the manuscript, in yellow.

  1. Thank you for your feedback. We have revised the abstract to improve its organization and continuity and have included a statement about the significance of our study.
  2. Thank you for your useful suggestions, we have now added a brief description of how we selected information from the literature and the doses that were used, in the “Results and discussion” section, from lines 487 to 496.
  3. Thank you for your suggestion. We have expanded the information on the biomedical application of Norcantharidin nanosystems in the Results and Discussion section from lines 330 to 335, and from lines 506 to 519, highlighting its potential in vitro applications, discussing the main experimental conditions used in previous studies.
  4. Thank you for your comment. We have now included a more detailed description of the mechanism of action of Norcantharidin in the “Results and discussion” section, from lines 571 to 580, complementing the information with a reference that you kindly suggested. We have also highlighted the potential of Norcantharidin as a therapeutic agent for melanoma. We hope that this addresses your concern adequately.
  5. Thank you for bringing this to our attention. We have revised the manuscript to include a discussion on the potential of the nanoemulsion of Norcantharidin as a solid alternative or a source of new drugs for melanoma treatment. We have included our findings and taking into consideration the previous studies that you suggested to support our discussion.
  6. Thank you for your valuable comment. We have discussed the possible toxicity of the proposed nanomaterials and the mitochondria targeting using this system in the “Results and discussion” section, from lines 541 to 556.

The “Conclusion” section has also been modified and improved.

We hope that these revisions adequately address your concerns and improve the overall quality of the manuscript. Thank you once again for your valuable feedback.

Reviewer 3 Report

The authors attempted the formulation of norcantharidin nanoemulsion for melanoma treatment. The following needs to be addressed:

1- The introduction section needs to include the previously published results on the anticancer activity of norcantharidin for melanoma treatment in particular.

2- Since norcantharidin is exposed to a temperature of 70 celsius, did the authors check its thermal stability?

3- For the in vitro release experiment, why did the authors use buffer for the receptor compartment? this way the release would occur via non-sink conditions

4- Figure 2 needs to include S.D., and the values need to be compared statistically

5- Generally throughout the manuscript and figures, values need to be reported as mean and S.D.

6- The conclusions section needs to be significantly reduced. The limitations and future perspective also need to be included in the manuscript.

Author Response

Thank you for your comments and suggestions regarding our manuscript on the formulation of norcantharidin nanoemulsion for melanoma treatment. A point-by-point answer is given bellow, and changes have been marked in the revised version of the manuscript, in yellow.

  1. Thank you for your valuable feedback. We have incorporated previously published results on the anticancer activity of norcantharidin in the introduction section from lines 91 to 102.

  1. Thank you for your comment. Regarding the thermal stability of norcantharidin, we appreciate your suggestion and have rephrased lines 138 to -141 to improve the clarity of the text. NCTD is added when the temperature drops to 40°C to prevent thermolabile degradation.

  1. We thank the reviewer for their comment. We used buffer for the receptor compartment in the in vitro release experiment to mimic the physiological pH conditions of the skin. Additionally, we maintained sink conditions to comply with the recommended technique by the European Pharmacopoeia for Franz cells experiments.

4 and 5. Thank you for the valuable insight. We acknowledge your suggestion to include standard deviation (S.D.) and perform statistical comparison of the values in Figure 2. As mentioned in line 337, we have updated the figure accordingly and have performed the necessary statistical analysis. However, we would like to clarify that Figures 3, 4, and 5 are in a format that are meant to be a graphical description of the assay and do not require the inclusion of S.D. or Standard Error. Furthermore, in Figure 6, we have only represented the percentage of drug released, and the corresponding S.D. and Standard Error are detailed in Table 2. Regarding Figure 7, we appreciate your interest in the methodology used to generate the plot. As mentioned in the manuscript, the plot was obtained using the Language and Environment for Statistical Computing 2022: R. Regarding Figure 8, we would like to confirm that the experiments were performed independently and were compared using the ANOVA statistical assay.

  1. We appreciate your comment and have significantly reduced the conclusions section while including a more detailed discussion of the limitations and future perspectives of our study.

Thank you again for your valuable feedback, which has helped us improve the manuscript.